# Efficacy of *Clostridium butyricum* Supplementation Combined with Phototherapy for Neonatal Hyperbilirubinemia: A Systematic Review and Meta-Analysis

**DOI:** 10.3390/microorganisms13071441

**Published:** 2025-06-20

**Authors:** Eun-Jin Kim, Ho-Yeon Go, Hyun-Kyung Sung

**Affiliations:** 1Department of Pediatrics of Korean Medicine, Korean Medicine Hospital, Dongguk University Bundang Medical Center, Seongnam-si 13601, Republic of Korea; utopialimpid@naver.com; 2Department of Korean Internal Medicine, College of Korean Medicine, Semyung University, Jecheon-si 27136, Republic of Korea; 3Department of Education, College of Korean Medicine, Dongguk University, Gyeongju-si 38066, Republic of Korea

**Keywords:** *Clostridium butyricum*, probiotics, phototherapy, neonatal hyperbilirubinemia

## Abstract

Neonatal hyperbilirubinemia (NH), which commonly presents as jaundice, affects approximately 60% of term infants and up to 80% of preterm infants within the first week of life. This study aimed to assess the efficacy and safety of *Clostridium butyricum* (*C. butyricum*) supplementation combined with phototherapy versus phototherapy alone for the treatment of NH. A systematic search of 11 databases (English, Chinese, and Korean) was conducted from 18 April 2025. Eligible randomized controlled trials (RCTs) compared *C. butyricum* plus phototherapy with phototherapy alone. Meta-analyses were performed using the mean difference (MD), standardized mean difference (SMD), and risk ratio (RR) with a 95% confidence interval (CIs). Evidence quality was evaluated using the GRADE approach. This review included 20 RCTs of 1054 neonates. Compared to phototherapy alone, *C. butyricum* supplementation significantly reduced total bilirubin (SMD = −1.54, 95% CI: −2.21 to −0.86), indirect bilirubin (SMD = −2.03, 95% CI: −2.98 to −1.07), and time to jaundice resolution (MD = −1.20 days, 95% CI: −1.66 to −0.75), and was associated with fewer adverse events (RR = 0.40, 95% CI: 0.30 to 0.55) (all *p* < 0.0001). These findings suggest that *C. butyricum* may have potential as a supportive adjunct therapy for neonatal hyperbilirubinemia. However, further studies are needed to confirm its efficacy and safety. The protocol is registered in PROSPERO (CRD420251031376).

## 1. Introduction

Neonatal hyperbilirubinemia (NH), commonly manifested as jaundice, is a prevalent condition in the early neonatal period, affecting approximately 60% of term and up to 80% of preterm infants within the first week of life [1]. It primarily results from the accumulation of unconjugated bilirubin owing to immature hepatic metabolism. If not promptly and appropriately managed, elevated bilirubin levels can cross the blood–brain barrier, potentially causing neurotoxicity and leading to severe complications, such as kernicterus. Phototherapy is a standard treatment because it facilitates the conversion of bilirubin into water-soluble isomers that can be excreted without conjugation [2]. Although phototherapy is effective, it is often associated with prolonged hospitalization and potential side effects, including dehydration, electrolyte imbalance, and skin-related complications, such as bronze baby syndrome. These limitations have prompted an increasing interest in adjunctive strategies that may accelerate bilirubin clearance and improve clinical outcomes.

One strategy involves targeting the gut microbiota, which plays a key role in the metabolism of bilirubin. Neonates are born with a nearly sterile gut, and delayed microbial colonization may impair bilirubin elimination by increasing intestinal β-glucuronidase activity and enhancing enterohepatic recirculation of bilirubin [3]. Probiotics have been proposed as potential adjunct therapies for addressing this problem. Their therapeutic benefits are believed to derive from the regulation of gut microbiota composition, reduction of intestinal pH, suppression of β-glucuronidase activity, enhancement of intestinal motility, and improvement in feeding tolerance, all contributing to improved bilirubin excretion [4,5,6].

Several reviews have explored the clinical efficacy of probiotics for the management of NH. According to previous systematic reviews, *Bifidobacterium*, *Saccharomyces boulardii*, *Clostridium butyricum* (*C. butyricum*), probiotic oligosaccharides, and *Bacillus subtilis* have demonstrated beneficial effects, particularly when used in conjunction with phototherapy [7]. However, other reviews indicate that while probiotics may reduce the duration of phototherapy, the overall evidence remains limited in both quantity and quality, underscoring the need for further well-designed studies [8].

To date, numerous randomized controlled trials (RCTs) have investigated the clinical efficacy of probiotics in the management of NH with phototherapy. Among these, probiotic supplementation containing *C. butyricum* has been shown to support intestinal health by enhancing mucosal barrier function and maintaining microbial balance. These effects may promote the reduction and excretion of conjugated bilirubin, thereby contributing to lower serum bilirubin levels [9,10]. A recent meta-analysis [10] reported that *C. butyricum*-based dual probiotic capsules, when used in combination with phototherapy, were more effective than phototherapy alone and did not increase the incidence of adverse events, highlighting their potential for broader clinical applications.

In light of these findings and to update the existing evidence, this study aimed to systematically evaluate the therapeutic effectiveness and safety of *C. butyricum* supplementation in combination with phototherapy for the treatment of NH. By focusing on studies published from 2020 onward, the present review seeks to provide an up-to-date evidence base to inform clinical decision-making and support the integration of strain-specific probiotic therapies in NH.

## 2. Materials and Methods

### 2.1. Protocol and Registration

This systematic review and meta-analysis was conducted in accordance with the Preferred Reporting Items for Systematic Reviews and Meta-Analyses (PRISMA) guidelines [11] (see Appendix A). The review protocol was prospectively registered in PROSPERO (registration no. CRD420251031376) on 15 April 2025 and is publicly accessible at https://www.crd.york.ac.uk/PROSPERO/view/CRD420251031376 (accessed on 14 June 2025).

### 2.2. Eligibility Criteria

#### 2.2.1. Types of Studies

This systematic review and meta-analysis aimed to evaluate the efficacy of combining *C. butyricum* supplementation with phototherapy, compared to phototherapy alone, in NH. Studies were excluded if they satisfied any of the following criteria: non-RCTs, study protocols, animal experiments, case reports, academic theses, survey studies, or review articles.

#### 2.2.2. Types of Participants

The participants were neonates diagnosed with hyperbilirubinemia, including both physiological and pathological jaundice, according to the diagnostic and therapeutic guidelines for NH. Neonates were excluded if they had jaundice secondary to congenital disorders, heart disease, or other congenital malformations.

#### 2.2.3. Types of Interventions

The experimental group received *C. butyricum* supplementation in combination with phototherapy, including both dual- and triple-strain probiotic formulations containing *C. butyricum*.

#### 2.2.4. Types of Comparisons

The control group was treated with phototherapy alone without *C. butyricum* supplementation.

#### 2.2.5. Types of Outcome Measurements

The outcome measures evaluated in this review included primary and additional clinical endpoints. The primary outcomes were (1) serum bilirubin levels, including total bilirubin, direct bilirubin, and indirect bilirubin levels; (2) time to jaundice resolution (days); and (3) total effective rate (TER). Secondary outcomes included (1) incidence of adverse events, (2) transcutaneous jaundice index, and (3) length of hospital stay (days).

### 2.3. Information Sources and Search Strategy

A systematic and comprehensive search of 11 electronic databases was conducted for studies published on or after 1 January 2020. The search was completed on 18 April 2025. This timeframe was selected to capture the most recent evidence and reflect updates from studies published in the past 5 years. The databases searched included three international databases: MEDLINE (via PubMed), EMBASE, and the Cochrane Central Register of Controlled Trials (CENTRAL); three Chinese databases (China National Knowledge Infrastructure [CNKI], Chinese Scientific Journal Database [VIP], and WanFang Data); and five Korean databases (Oriental Medicine Advanced Searching Integrated System [OASIS], Korean Studies Information Service System [KISS], Korea Citation Index [KCI], Research Information Sharing Service [RISS], and Korean Medical Database [KMbase]). The search strategy used a combination of terms related to “neonatal hyperbilirubinemia” and “probiotics”, which were adapted for each database to reflect specific indexing terms and language conventions. A detailed account of the search terms and results for each database is provided in Appendix A.

### 2.4. Study Selection and Data Extraction

#### 2.4.1. Study Selection

Two authors (E.-J.K. and H.-K.S.) independently screened the titles and abstracts of all the retrieved records. Full-text articles were evaluated for eligibility based on the predefined inclusion and exclusion criteria. Discrepancies were resolved through mutual discussion between the three authors, and a consensus was achieved in all cases.

#### 2.4.2. Data Extraction

Two authors (E.-J.K. and H.-K.S.) independently performed data extraction using a standardized form. Discrepancies were resolved through discussion with all authors. In instances where data were missing or unclear, the corresponding authors were contacted via email for clarification. The extracted information included the first author’s name, year of publication, sample size, total treatment duration, participant characteristics, details of the interventions and comparators, outcome measures, adverse events, and data relevant to the assessment of bias (RoB).

### 2.5. Assessment of RoB

The RoB for each included study was independently evaluated by two authors (H.-Y.G. and H.-K.S.) using the Cochrane RoB 2 tool, in accordance with the recommendations outlined in the Cochrane Handbook for Systematic Reviews of Interventions [12]. Each study was classified as having a low risk, some concerns, or a high risk of bias. The assessment covered five key domains: (1) randomization process, (2) deviations from the intended interventions, (3) missing outcome data, (4) measurement of outcomes, and (5) selection of the reported results.

### 2.6. Statistical Analysis

All included studies were initially qualitatively synthesized. When two or more studies reported comparable outcome measures, whether continuous or dichotomous, a meta-analysis was conducted using Review Manager (RevMan) version 5.4 (Cochrane Collaboration, London, UK). For dichotomous outcomes, Risk ratios (RRs) with 95% confidence intervals (CIs) were calculated. For continuous outcomes, either mean differences (MDs) or standardized mean differences (SMDs) were used depending on the consistency of measurement units across studies, each reported with corresponding 95% CIs.

#### 2.6.1. Assessment of Heterogeneity

Heterogeneity across the included studies was assessed using Higgins I^2^ statistic [13]. An I^2^ value of 50% or higher indicated substantial heterogeneity, in which case a random-effects model was employed for data synthesis. Conversely, when the I^2^ value was <50%, heterogeneity was considered low, and a fixed-effects model was used.

#### 2.6.2. Assessment of Reporting Bias

Publication bias was assessed for outcomes reported in 10 or more studies. The funnel plots were visually inspected to detect potential asymmetries. When visual inspection suggested possible bias, additional statistical tests were conducted, including Egger’s regression test, Rosenthal’s fail-safe N test, and the trim-and-fill method. These analyses were conducted using R Studio (version 1.4.1106; R Studio, PBC, Boston, MA, USA) with the “meta” package and its default settings.

#### 2.6.3. Subgroup and Sensitivity Analysis

If significant heterogeneity was noted in the meta-analysis, subgroup analyses were conducted to explore potential sources of variability. Subgroup analyses were conducted based on the type of probiotic formulation (e.g., single-, dual-, or triple-strain *C. butyricum*) and the classification of jaundice (physiological vs. pathological), only when sufficient data were available to justify such comparisons. Sensitivity analyses were independently conducted for each outcome when more than 10 studies were included.

### 2.7. Quality of Evidence

The certainty of evidence was assessed using the Grading of Recommendations Assessment, Development, and Evaluation (GRADE) approach in accordance with the standardized criteria available at http://gradepro.org (accessed on 4 May 2025). The quality of evidence for each outcome was evaluated across five domains: risk of bias, inconsistency, indirectness, imprecision, and publication bias. Based on these domains, the overall certainty of the evidence was rated as high, moderate, low, or very low following the GRADE framework.

## 3. Results

### 3.1. Results of Literature Search

A total of 1054 records were identified through database searches comprising 45 English, 782 Chinese, and 227 Korean databases. After duplicates were removed, 661 unique records remained. Following title and abstract screening, 434 studies were excluded because they did not satisfy the initial eligibility criteria, and the full texts of three additional studies could not be retrieved. As a result, 224 articles were selected for a full-text review. Among them, 42 studies were excluded because they were not RCTs, and 162 were excluded because of inappropriate interventions that did not satisfy the inclusion criteria. Thus, 20 RCTs [14,15,16,17,18,19,20,21,22,23,24,25,26,27,28,29,30,31,32,33] were included in the final systematic review and meta-analysis (Figure 1).

### 3.2. Characteristics of the Study

All 20 included RCTs were conducted in China and were published between 2020 and 2024. The sample sizes ranged from 60 to 180 participants. The duration of treatment varied between 3 and 2 weeks. The age of neonates ranged from 2.36 ± 2.26 to 18.94 ± 1.06 days, and gestational age ranged from 38.58 ± 0.72 to 40.52 ± 1.09 weeks. Reported birth weights ranged from 3020.15 ± 1049.85 g to 4.30 ± 1.29 kg. One study [14] specifically reported birth weight, and three studies [17,28,29] provided data on the duration of illness, which ranged from 4.20 ± 0.23 to 6.41 ± 3.02 days.

Regarding jaundice classification, seven studies [19,21,25,27,28,30,33] explicitly identified this condition as pathological jaundice. Five studies [19,27,30,31,33] provided detailed subclassifications, including infectious, hemolytic, hepatocellular, cholestatic, perinatal-related, and breast milk jaundice. Conversely, the remaining studies broadly referred to this condition as NH without further specification.

As for diagnostic criteria, three studies [18,26,29] did not clearly describe the basis for diagnosis. The remaining studies enrolled participants with laboratory-confirmed hyperbilirubinemia, consistent with the diagnostic and therapeutic guidelines for NH (Table 1).

### 3.3. Interventions

Probiotics were administered orally in all included studies. *C. butyricum* has been used either as a single-strain probiotic or as part of a multi-strain formulation. Specifically, four studies [18,28,29,32] used *C. butyricum* alone, and one study [14] administered *C. butyricum* double-living capsules. In 13 studies [15,17,19,21,22,23,24,25,26,27,30,31,33], dual-strain formulations were employed, combining *C. butyricum* with *Bifidobacterium*. In two studies [16,20], triple-strain formulations were used, consisting of *C. butyricum*, *Enterococcus faecalis*, and *Bacillus mesentericus*. The detailed compositions, dosages, and administration frequencies of the probiotic interventions are summarized in Table 2. Additionally, all included studies incorporated phototherapy as part of the intervention, with the specific phototherapy protocols provided in Table 2.

### 3.4. Outcome Measures

Primary outcome measures included serum bilirubin levels (total, direct, and indirect) before and after treatment, time to jaundice resolution (days), and TER. Total bilirubin levels were reported in 15 studies [15,17,18,19,20,21,22,23,24,25,26,29,30,32,33]. Among these, two studies [18,20] provided only posttreatment values without corresponding pretreatment data and were therefore excluded from the analysis of mean changes. Moreover, two studies [23,33] were excluded because of suspected inconsistencies in the reported bilirubin data. One study [32] did not clearly specify whether the reported values referred to total, direct, or indirect bilirubin and was also excluded from the analysis. Consequently, 10 studies [15,17,19,21,22,24,25,26,29,30] were included in the meta-analysis of total bilirubin levels. Direct bilirubin levels were assessed in eight studies [15,17,21,23,24,25,29,33]. Four studies [17,23,29,33] were excluded owing to suspected data errors. Consequently, four studies [15,21,24,25] were included in the meta-analysis of direct bilirubin levels. Indirect bilirubin levels were reported in 11 studies [15,17,18,19,22,23,24,25,26,29,33]. One study [18] was excluded because it reported only posttreatment values, preventing a comparison of pre- and post-changes. Moreover, four studies [17,23,29,33] were excluded because of suspected errors in the reported indirect bilirubin values. A total of six studies [15,19,22,24,25,26] were included in the meta-analysis. In addition, two studies [32,33] reported only the average daily reduction in bilirubin levels, and one study [31] reported the time required for bilirubin levels to return to normal. Therefore, these studies were excluded from the meta-analysis. Time to jaundice resolution was evaluated in six studies [16,21,25,30,32,33], while TER was reported in 18 studies [14,15,16,17,18,19,20,21,23,24,26,27,28,29,30,31,32,33].

Secondary outcome measures included adverse events, transcutaneous jaundice index (measured before and after treatment), and length of hospital stay (days). Adverse events were assessed in 19 studies [14,15,16,17,19,20,21,22,23,24,25,26,27,28,29,30,31,32,33] (Appendix A), while the transcutaneous jaundice index was evaluated in five studies [14,16,21,26,30], and the length of hospital stay was reported in four studies [20,21,24,31]. Comprehensive details of all secondary outcome measures and their corresponding *p*-values are presented in Appendix A.

### 3.5. Quality Assessment

RoB was assessed using the Cochrane RoB 2 tool. All included studies were rated as having some concerns in the domain of the randomization process owing to insufficient information on allocation sequence concealment. There were no deviations from the intended interventions, and all studies were deemed to have a low risk in this domain. Similarly, all studies were assessed to have a low risk of bias in the domain of missing outcome data. Most studies had a low risk of bias in the outcome measurements. However, one study [14] raised concerns because of missing units for indicators, limiting interpretability. Additionally, four studies [17,23,29,33] reported bilirubin levels outside physiologically plausible ranges without justification, leading to concerns regarding outcome measurements. In the domain of selective reporting, most studies were assessed as low risk, but one study [26] showed a discrepancy in the reported sample size, raising some concerns. Overall, six studies [14,17,23,26,29,33] were rated as having a high risk of bias, while the remaining studies were judged to have some concerns. These evaluations are shown in Figure 2 and Figure 3.

### 3.6. Meta-Analysis Results

A meta-analysis was conducted to compare the efficacy of probiotics combined with phototherapy to that of phototherapy alone for the management of NH.

#### 3.6.1. Serum Bilirubin Level

Serum bilirubin levels before and after treatment were analyzed using SMD to account for differences in baseline values and treatment durations across the included studies.

Ten studies [15,17,19,21,22,24,25,26,29,30], involving 870 neonates, reported total bilirubin levels before and after treatment and were included in the meta-analysis. A meta-analysis evaluating the effect of probiotics combined with phototherapy on total bilirubin levels demonstrated a significant reduction compared to phototherapy alone. The pooled SMD was −1.54 (95% CI: −2.21 to −0.86), indicating a considerable treatment effect. Although substantial heterogeneity was noted across the studies (I^2^ = 95%, *p* < 0.00001), the overall effect remained statistically significant (Z = 4.47, *p* < 0.00001), suggesting that probiotics may offer beneficial adjunctive effects in NP management (Figure 4A).

Four studies [15,21,24,25], involving 410 neonates, reported direct bilirubin levels before and after treatment and were included in the meta-analysis. The pooled SMD was −0.91 (95% CI: −1.32 to −0.50), indicating a statistically significant reduction in direct bilirubin levels following the intervention. Substantial heterogeneity was detected across the studies (I^2^ = 73%, *p* = 0.01), prompting the use of a random-effects model. Despite the high heterogeneity, the overall treatment effect remained statistically significant (Z = 4.34, *p* < 0.0001), supporting the potential utility of probiotics as an effective adjunct to phototherapy for managing NH (Figure 4B).

A total of six studies [15,19,22,24,25,26], involving 448 neonates, reported indirect bilirubin levels before and after treatment and were included in the meta-analysis. The pooled SMD was −2.03 (95% CI: −2.98 to −1.07), suggesting a trend toward reduction in indirect bilirubin levels following treatment with probiotics combined with phototherapy (Z = 4.17, *p* < 0.0001). Substantial heterogeneity was detected among the included studies (I^2^ = 94%, *p* < 0.00001), prompting the use of a random-effects model (Figure 4C).

#### 3.6.2. Duration Until Jaundice Resolution

Seven studies [16,21,25,30,32,33], involving 636 neonates, reported the duration required for jaundice resolution (in days) and were included in the meta-analysis. The pooled MD was −1.20 days (95% CI: −1.66 to −0.75), indicating that the addition of probiotics to phototherapy significantly shortened the time needed for jaundice to resolve compared to phototherapy alone. Substantial heterogeneity was detected among the included studies (I^2^ = 95%, *p* < 0.00001), prompting the use of a random effects model. Despite this high level of variability, the overall effect remained highly significant (Z = 5.15, *p* < 0.00001), indicating the consistent benefit of the combined treatment in accelerating jaundice clearance in neonates (Figure 5).

#### 3.6.3. TER

Eighteen studies [14,15,16,17,18,19,20,21,23,24,26,27,28,29,30,31,32,33] involving 1679 neonates assessed the TER of probiotics combined with phototherapy compared with phototherapy alone. The pooled analysis demonstrated an RR of 1.20 (95% CI: 1.15 to 1.25), indicating that the intervention group exhibited a significantly higher rate of clinical improvement than the control group. Moderate heterogeneity was observed across the studies (I^2^ = 45%, *p* = 0.02); thus, a fixed-effects model was used. The overall effect was highly significant (Z = 9.21, *p* < 0.00001), supporting the consistent and favorable impact of probiotics as an adjunct therapy to phototherapy in the treatment of NH (Figure 6).

#### 3.6.4. Adverse Events

A total of 19 studies [14,15,16,17,19,20,21,22,23,24,25,26,27,28,29,30,31,32,33], involving 1637 neonates, evaluated the incidence of adverse events associated with probiotic supplementation combined with phototherapy. The meta-analysis yielded a pooled RR of 0.40 (95% CI: 0.30 to 0.55), indicating a significantly lower risk of adverse events in the intervention group compared to the control group. No heterogeneity was noted across the included studies (I^2^ = 0%, *p* = 0.71), supporting the application of the fixed-effects model. The test for overall effect was highly significant (Z = 5.77, *p* < 0.00001), indicating that the addition of probiotics to phototherapy is not only effective but also confers a favorable safety profile in the treatment of NH (Figure 7).

#### 3.6.5. Transcutaneous Bilirubin Level

A meta-analysis of five studies [14,16,21,26,30], including 502 neonates, evaluated the changes in transcutaneous bilirubin levels before and after treatment. Due to differences in the reporting units (e.g., mg/dL and µmol/L), all values were converted to µmol/L, and the analysis was conducted using the SMD. The pooled SMD was −1.11 (95% CI: −1.62 to −0.59), indicating a statistically significant reduction in jaundice levels following the intervention. Substantial heterogeneity was observed (I^2^ = 85%, *p* < 0.0001), and a random effects model was used. The overall effect was highly significant (Z = 4.23, *p* < 0.0001), supporting the efficacy of probiotic supplementation combined with phototherapy in reducing transcutaneous bilirubin levels in neonates (Figure 8).

#### 3.6.6. Length of Hospital Stay

A meta-analysis of four studies [20,21,25,31], including a total of 446 neonates, evaluated the impact of probiotics combined with phototherapy on the length of hospital stay in the management of NH. The pooled analysis revealed a significant MD of −1.66 days (95% CI: −2.44 to −0.88), indicating a shorter duration of hospitalization in the intervention group. Substantial heterogeneity was noted among the included studies (I^2^ = 94%, *p* < 0.00001); therefore, a random effects model was employed. The overall effect was highly significant (Z = 4.18, *p* < 0.0001), supporting the potential benefit of adjunct probiotic therapy in reducing the hospital stay of neonates undergoing phototherapy (Figure 9).

### 3.7. Publication Bias

The funnel plot for total bilirubin levels suggested potential publication bias, which was further evaluated using the R Studio program (Version 1.4.1106, Integrated Development for R, R Studio, PBC, Boston, MA, USA) with the ‘meta’ package as the default setting. Egger’s test indicated a statistically significant asymmetry in the funnel plot (t = −2.82, df = 8, *p* = 0.0225), indicating the possibility of publication bias (Figure 10). The trim-and-fill analysis imputed one potentially missing study, and after adjustment, the overall effect size slightly decreased from SMD −1.32 to −1.32 (95% CI: −2.08, −0.55), while maintaining statistical significance (Z = −3.37, *p* = 0.0008). Furthermore, the fail-safe N was calculated to be 1120, indicating that more than a thousand null-result studies are needed to nullify the observed effect. Collectively, despite indications of publication bias, the treatment effects remained statistically significant and reliable.

### 3.8. Sensitivity Analyses and Subgroup Analyses

A subgroup analysis was considered to explore potential sources of heterogeneity based on the type of probiotic formulation and the classification of jaundice. However, it was not conducted due to the insufficient number of studies within each subgroup, which could have resulted in misleading or inconclusive findings.

Sensitivity analysis, conducted by sequentially excluding each study, demonstrated that the overall effect on total bilirubin levels remained statistically significant (SMD range: −1.31 to −1.69, Appendix A), indicating the robustness and stability of the meta-analysis results despite high heterogeneity (I^2^ = 92–95%, all *p* < 0.00001).

### 3.9. GRADE Certainty of Evidence

The certainty of the evidence was assessed using the GRADE approach, and the results are summarized in Table 3. The outcomes for the total bilirubin level, direct bilirubin level, indirect bilirubin level, time to jaundice fading, total effective rate, and adverse events were rated with moderate certainty. In contrast, the certainty of evidence for transcutaneous bilirubin levels and the length of hospital stay was low. The detailed reasons for downgrading the quality of evidence for each outcome are presented in Table 3.

## 4. Discussion

### 4.1. Summary of This Review

This systematic review and meta-analysis evaluated the efficacy and safety of probiotics in combination with phototherapy compared with phototherapy alone in the management of NH. After a comprehensive literature search, 20 RCTs involving 1715 neonates were included. The combined use of *C. butyricum* supplementation and phototherapy has demonstrated significant clinical advantages over phototherapy alone, including accelerated resolution of jaundice, reduced transcutaneous bilirubin levels, shorter hospital stays, and higher TER. Specifically, notable improvements were observed in the reduction of serum bilirubin levels. The pooled SMDs were −1.54 (95% CI: −2.21 to −0.86) for total bilirubin, −0.91 (95% CI: −1.32 to −0.50) for direct bilirubin, and −2.03 (95% CI: −2.98 to −1.07) for indirect bilirubin. These results indicate a particularly marked reduction in indirect bilirubin levels, which is a key target of phototherapy and an important marker for evaluating its efficacy. Furthermore, *C. butyricum* supplementation lowers the incidence of adverse effects, such as fever, rash, diarrhea, and dehydration. These findings suggest that *C. butyricum* supplementation may serve as an effective and safe adjunctive therapy to enhance phototherapy outcomes in neonates with hyperbilirubinemia.

### 4.2. Clinical Implications, Limitations, and Suggestions

Epidemiological studies suggest that neonatal jaundice occurs more frequently in East Asian populations compared to other ethnic groups. Specifically, approximately 49% of East Asian neonates develop clinically significant hyperbilirubinemia compared to 20% of White and 12% of Black infants [34]. These findings highlight the need for effective adjunct interventions to enhance bilirubin clearance, particularly in vulnerable populations.

The risk of hyperbilirubinemia in neonates is primarily attributed to the immaturity of both their hematologic and hepatic systems. Neonatal red blood cells are larger and have a shorter lifespan (60–90 days in term infants; 35–50 days in preterm infants) compared to adult cells, leading to increased hemolysis and bilirubin production [35]. Additionally, reduced activity of uridine diphosphate glucuronosyltransferase 1A1 (UGT1A1)—the key enzyme responsible for bilirubin conjugation—results in impaired clearance. In addition, neonates with glucose-6-phosphate dehydrogenase (G6PD) deficiency have an increased risk of hyperbilirubinemia [36]. The American Academy of Pediatrics recommends considering G6PD deficiency as a high-risk factor for neonatal jaundice in infants ≥35 weeks of gestation [37].

The concept of a bilirubin–intestinal microbiota–NH cycle provides a novel perspective on the underlying mechanisms of neonatal jaundice. The neonatal gut microbiome, which begins to establish shortly after birth, plays a crucial role in regulating bilirubin metabolism. Delayed or insufficient colonization by beneficial gut bacteria may lead to increased enterohepatic circulation of bilirubin, thereby elevating serum bilirubin levels. To address this, probiotic supplementation has been proposed as a potential therapeutic strategy to modulate the gut microbiota—by promoting early colonization, enhancing intestinal motility, and reducing β-glucuronidase activity, all of which support bilirubin clearance [3]. While previous studies suggest a potential role of probiotics in managing neonatal hyperbilirubinemia, the current evidence remains limited and should be interpreted with caution. Therefore, this study was conducted to evaluate the efficacy and safety of probiotic formulations as an adjunctive treatment in this population.

*C. butyricum* is widely used in East Asia, including Japan, Korea, and China, in the form of nontoxigenic strains with a favorable safety profile [38]. In pediatrics, current research on *C. butyricum* has been associated with improved clinical outcomes, such as reduced duration of diarrhea, enhanced weight gain, and prevention of antibiotic-associated diarrhea [39]. It has also demonstrated therapeutic potential in managing persistent gastrointestinal disorders, such as inflammatory bowel disease and irritable bowel syndrome [40,41], as well as in promoting feeding tolerance and growth in neonates receiving intensive care [42]. Furthermore, *C. butyricum* has protective effects against necrotizing enterocolitis and its ability to enhance gut barrier integrity [43]. These probiotic strains have the ability to restore microbial balance, reduce intestinal inflammation [3], and produce short-chain fatty acids that strengthen the intestinal barrier [39,41]. Taken together with previous studies, the present findings suggest that *C. butyricum* may serve as a valuable adjunctive therapy in the management of NH alongside phototherapy, particularly in high-risk populations such as East Asian infants.

This study had several limitations. Although an extensive search was conducted across databases without regional or language restrictions, all included RCTs were conducted in China, which may limit the generalizability of the findings. Due to the high incidence of neonatal jaundice among East Asian populations, a large number of related RCTs have been conducted in this region. The composition of the gut microbiota in infants is influenced by various factors, including mode of delivery, breastfeeding status, and ethnicity [44]. As this study is based on clinical trials conducted exclusively in China, the results should be interpreted with caution. Future clinical research should consider ethnic backgrounds to enhance the accuracy and relevance of treatment strategies.

Additionally, the certainty of evidence, as assessed by GRADE, ranged from moderate to very low. The number of studies that could be included in each meta-analysis was limited due to the use of diverse outcome measures across the included trials. Furthermore, some studies [17,23,29,33] were excluded from the bilirubin outcome analysis due to suspected reporting errors or data inconsistencies. Substantial heterogeneity was observed in several pooled outcomes (I^2^ > 90%), which may be partly explained by differences in phototherapy protocols, probiotic dosage or frequency, jaundice type, and the wide age range of enrolled neonates. Although subgroup analyses were initially planned to examine differential effects based on jaundice classification and probiotic formulation, they could not be performed due to insufficient data within each subgroup. Only seven studies [19,21,25,27,28,30,33] clearly identified cases of pathological jaundice; one study [31] included both pathological and physiological jaundice without distinction, while the remaining studies did not specify the type of jaundice. This lack of information limits the feasibility of subgroup evaluation by jaundice classification. Similarly, subgroup analysis by probiotic formulation was not possible because of the small number of studies in each category.

Our findings suggest the potential clinical utility of *C. butyricum* supplementation as a complementary strategy in the management of NH. This study may serve as a foundation for future investigations into the synergistic effects of probiotics combined with phototherapy. To validate these results, high-quality RCTs with standardized intervention protocols are warranted. Additionally, further analyses stratified by probiotic formulation type are necessary to better understand the potential interactions between phototherapy and various probiotic strains.

## 5. Conclusions

*C. butyricum* supplementation may serve as an adjunctive therapeutic option for the management of NH when combined with phototherapy. However, since this meta-analysis is based solely on studies conducted in China, the findings may have limited external validity and should therefore be interpreted with caution. Further research is warranted to elucidate the underlying mechanisms of probiotic action on bilirubin metabolism and to consider ethnic variations in gut microbiota that may impact therapeutic efficacy.

## Figures and Tables

**Figure 1 microorganisms-13-01441-f001:**
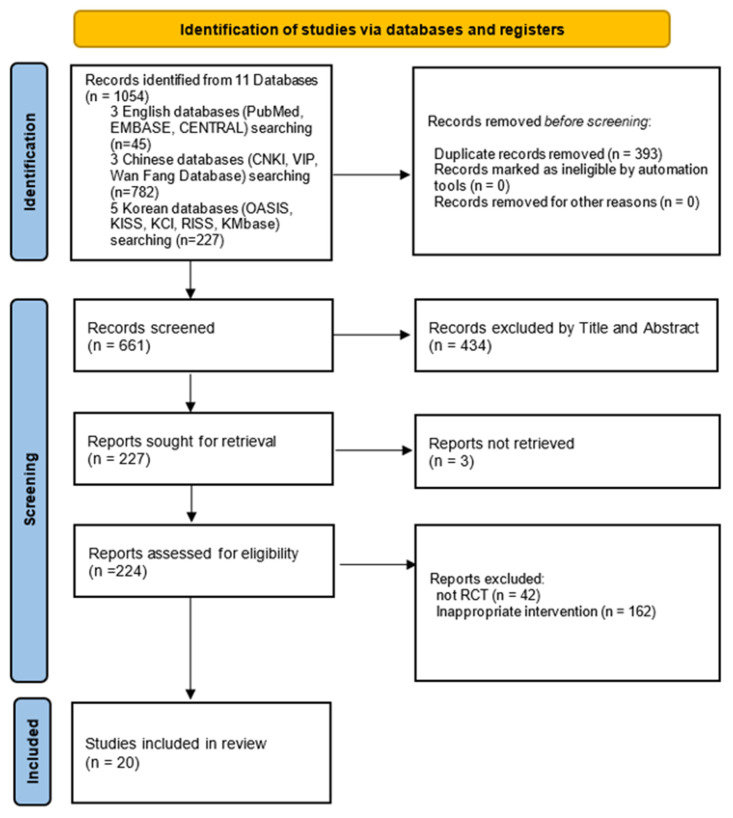
PRISMA flow diagram of study selection process. EMBASE, Excerpta Medica Database; CENTRAL, Cochrane Central Register of Controlled Trials; CNKI, China National Knowledge Infrastructure; VIP, Chinese Scientific Journal Database; OASIS, Oriental Medicine Advanced Searching Integrated System; KISS, Korean Studies Information Service System; KCI, Korea Citation Index; RISS, Research Information Sharing Service; KMbase, Korean Medical Database.

**Figure 2 microorganisms-13-01441-f002:**
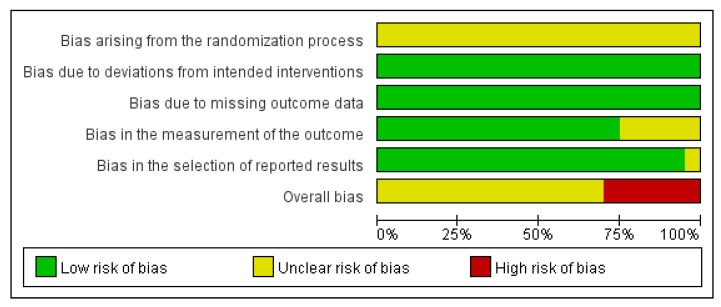
Risk of bias graph.

**Figure 3 microorganisms-13-01441-f003:**
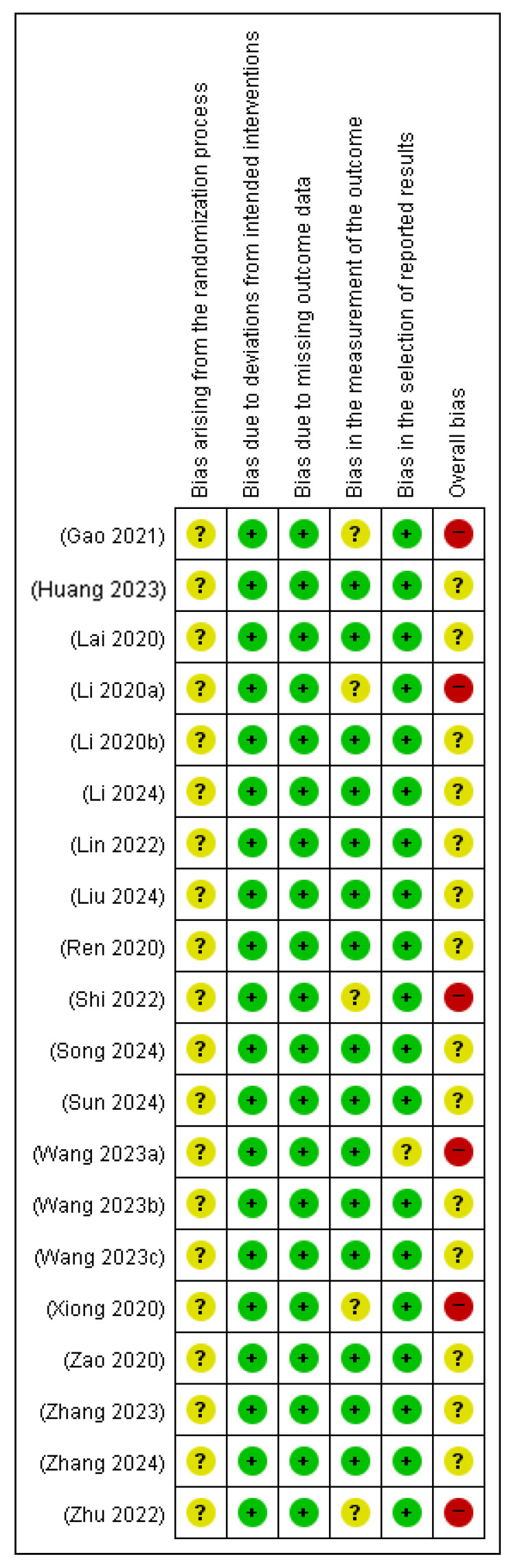
Risk of bias summary [14,15,16,17,18,19,20,21,22,23,24,25,26,27,28,29,30,31,32,33]. +, Low risk of bias; ?, Unclear risk of bias; −, High risk of bias.

**Figure 4 microorganisms-13-01441-f004:**
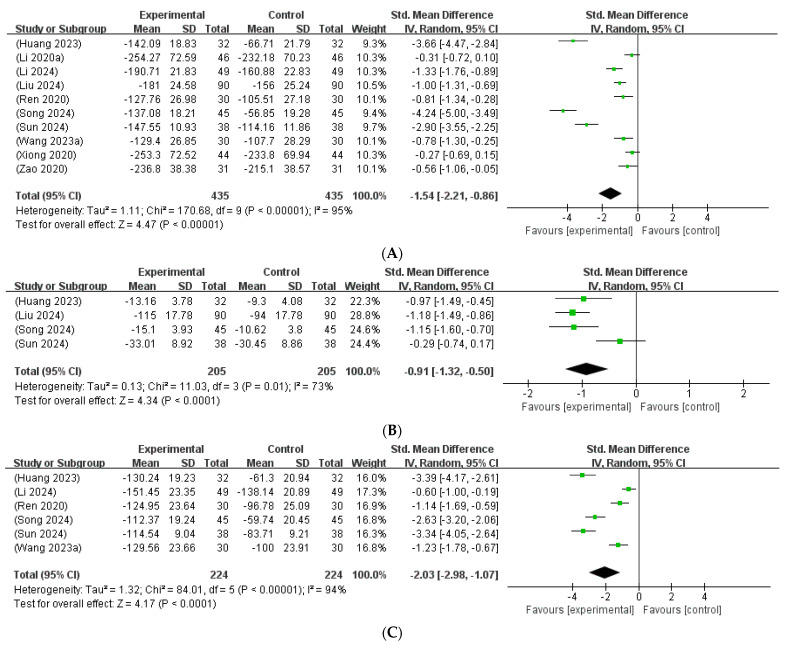
(**A**) Forest plot of the total bilirubin levels [15,17,19,21,22,24,25,26,29,30]. (**B**) Forest plot of the direct bilirubin levels [15,21,24,25]. (**C**) Forest plot of the indirect bilirubin levels [15,19,22,24,25,26].

**Figure 5 microorganisms-13-01441-f005:**
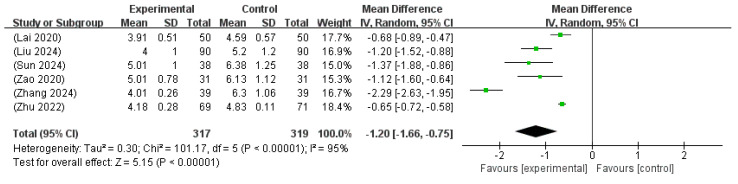
Forest plot of the time of resolution of jaundice [16,21,25,30,32,33].

**Figure 6 microorganisms-13-01441-f006:**
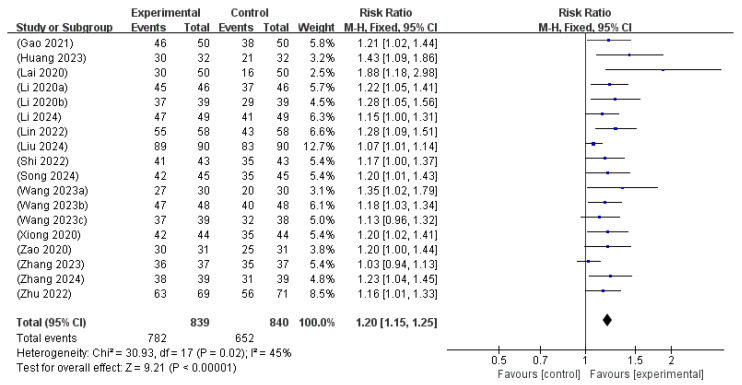
Forest plot of the total effective rate [14,15,16,17,18,19,20,21,23,24,26,27,28,29,30,31,32,33].

**Figure 7 microorganisms-13-01441-f007:**
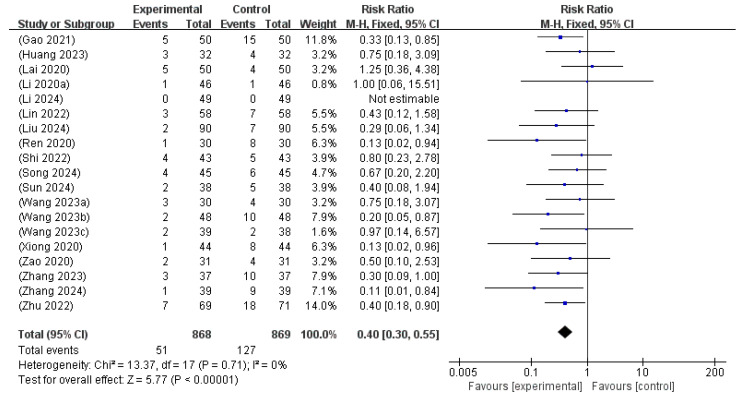
Forest plot of the adverse events [14,15,16,17,19,20,21,22,23,24,25,26,27,28,29,30,31,32,33].

**Figure 8 microorganisms-13-01441-f008:**
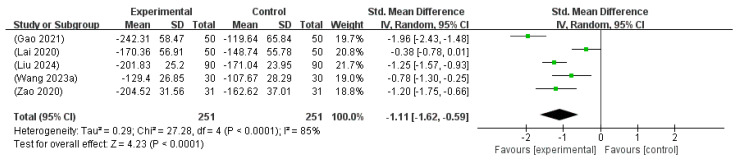
Forest plot of the transcutaneous jaundice levels [14,16,21,26,30].

**Figure 9 microorganisms-13-01441-f009:**
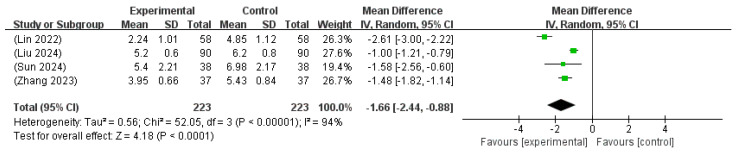
Forest plot of the length of hospital stay [20,21,25,31].

**Figure 10 microorganisms-13-01441-f010:**
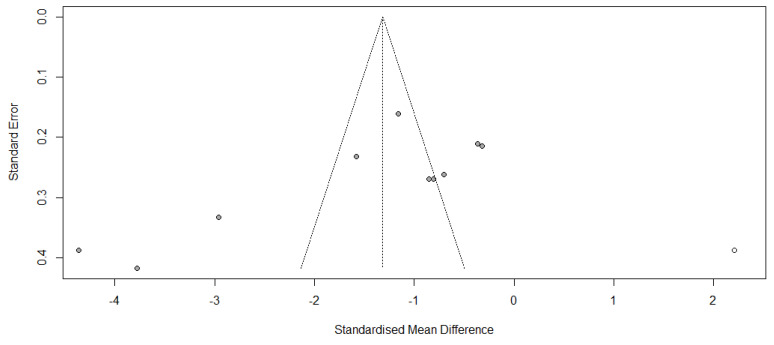
Adjusted funnel plot of the total bilirubin levels. The closed dots are real studies, and the open dots are filled artificial studies.

**Table 1 microorganisms-13-01441-t001:** Basic characteristics of the included studies.

First Author (Year)	Sample Size (E/C)	Age Distribution (Mean ± SD)	Body Weight [Birth Weight] (Mean ± SD)	Gestational Age (mean ± SD)	Duration of Illness (Mean ± SD)	Type of Jaundice	Experimental Intervention (E)	Total Treatment Periods	Outcome Measurement	Adverse Events Incidence
							Control Intervention (C)			
Gao (2021) [14]	100 (50/50)	E: 3.58 ± 1.291 d	[E: 3.39 ± 0.475 kg]	NR	NR	NR	Probiotics + (C)	3 d	(1)(2)(3)	E: 5/50
		C: 3.25 ± 1.401 d	[C: 3.28 ± 0.451 kg]	NR	NR	NR	phototherapy			C: 14/50
Huang (2023) [15]	64 (32/32)	E: 16.57 ± 3.11 d	NR	NR	NR	NR	Probiotics + (C)	5 d	(1)(2)(4)(5)(6)(7)(8)	E: 3/32
		C: 17.26 ± 3.52 d	NR	NR	NR	NR	phototherapy			C: 4/32
Lai (2020) [16]	100 (50/50)	E: 2~6 d	normal range	full-term infant	NR	NR	Probiotics + (C)	5 d	(1)(2)(3)(9)	E: 5/50
		C: 2~6 d	normal range	full-term infant	NR	NR	phototherapy	<5 d		C: 4/50
Li (2020a) [17]	92 (46/46)	E: 7.12 ± 2.31 d	E: 3.25 ± 0.21 kg	E: 38.75 ± 1.23 w	4.23 ± 1.16 d	NR	Probiotics + (C)	5 d	(1)(2)(4)	E: 1/46
		C: 7.15 ± 2.28 d	C: 3.24 ± 0.23 kg	C: 38.65 ± 1.05 w	4.28 ± 1.18 d	NR	phototherapy			C: 1/46
Li (2020b) [18]	78 (39/39)	E: 18.94 ± 1.06 d	NR	NR	NR	NR	Probiotics + (C)	5 d	(1)(4)	NR
		C: 18.22 ± 1.37 d	NR	NR	NR	NR	phototherapy			NR
Li (2024) [19]	98 (49/49)	E: 10.43 ± 2.02 d	E: 3.27 ± 0.33 kg	NR	NR	infectious (n = 25), hemolytic (n = 13), others (n = 11)	Probiotics + (C)	2 w	(1)(2)(4)(7)(10)(11)	E: 0/49
		C: 10.85 ± 2.07 d	C: 3.18 ± 0.35 kg	NR	NR	infectious (n = 24), hemolytic (n = 15), others (n = 10)	phototherapy			C: 0/49
Lin (2022) [20]	116 (58/58)	NR	NR	NR	NR	NR	Probiotics + (C)	NR	(1)(2)(4)(8)(12)(13)(14)(15)	E: 3/58
		NR	NR	NR	NR	NR	phototherapy			C: 7/58
Liu (2024) [21]	180 (90/90)	E: 5.2 ± 0.4 d	NR	NR	NR	pathological jaundice	Probiotics + (C)	7 d	(1)(2)(3)(4)(9)(15)(16)(17)	E: 2/90
		C: 5.2 ± 0.5 d	NR	NR	NR		phototherapy			C: 7/90
Ren (2020) [22]	60 (30/30)	E: 15.02 ± 6.54 d	E: 4.25 ± 1.20 kg	NR	NR	NR	Probiotics + (C)	5 d	(2)(4)	E: 1/30
		C: 15.01 ± 6.99 d	C: 4.30 ± 1.29 kg	NR	NR	NR	phototherapy			C: 8/30
Shi (2022) [23]	86 (43/43)	E: 7.05 ± 0.75 d	NR	E: 39.63 ± 0.52 w	NR	NR	Probiotics + (C)	5 d	(1)(2)(4)(18)	E: 4/30
		C: 5.52 ± 0.63 d	NR	C: 39.63 ± 0.52 w	NR	NR	phototherapy			C: 5/30
Song (2024) [24]	90 (45/45)	E: 5.24 ± 1.05 d	NR	E: 38.58 ± 0.72 w	NR	NR	Probiotics + (C)	7 d	(1)(2)(4)(8)(19)(20)	E: 4/45
		C: 5.29 ± 1.08 d	NR	C: 38.75 ± 0.79 w	NR	NR	phototherapy			C: 6/45
Sun (2024) [25]	76 (38/38)	E: 12.56 ± 3.18 d	NR	E: 39.39 ± 1.23 w	NR	pathological jaundice	Probiotics + (C)	5 d	(2)(4)(9)(15)(16)(17)	E: 2/45
		C: 12.56 ± 2.93 d	NR	C: 39.59 ± 1.26 w	NR		phototherapy			C: 5/45
Wang (2023a) [26]	60 (30/30)	E: 5.37 ± 0.22 d	NR	NR	NR	NR	Probiotics + (C)	8 d	(1)(2)(3)(4)	E: 3/30
		C: 5.41 ± 0.23 d	NR	NR	NR	NR	phototherapy			C: 4/30
Wang (2023b) [27]	96 (48/48)	E: 12.56 ± 3.63 d	E: 3.24 ± 0.68 kg	E: 40.51 ± 0.49 w	NR	infectious (n = 28), hemolytic (n = 13), perinatal factors (n = 4), others (n = 3)	Probiotics + (C)	2 w	(1)(2)(18)(21)(22) (23)(24)(25)(26)	E: 2/48
		C: 12.94 ± 3.81 d	C: 3.28 ± 0.73 kg	C: 40.38 ± 0.69 w	NR	infectious (n = 29), hemolytic (n = 11), perinatal factors (n = 5), others (n = 3)	phototherapy			C: 9/48
Wang (2023c) [28]	77 (39/38)	E: 8.04 ± 1.99 d	E: 3.26 ± 0.68 kg	E: 39.35 ± 1.87 w	6.41 ± 3.02 d	Pathological jaundice	Probiotics + (C)	3 d	(1)(2)(27)(28)	E: 2/39
		C: 7.97 ± 2.36 d	C: 3.24 ± 0.61 kg	C: 39.42 ± 1.63 w	6.29 ± 2.87 d		phototherapy			C: 2/38
Xiong (2020) [29]	88 (44/44)	E: 6.25 ± 1.46 d	NR	NR	4.23 ± 0.24 d	NR	Probiotics + (C)	4 d	(1)(2)(4)	E: 1/44
		C: 6.29 ± 1.51 d	NR	NR	4.20 ± 0.23 d	NR	phototherapy			C: 8/44
Zhao (2020) [30]	62 (31/31)	E: 13.86 ± 7.67 d	E: 3.45 ± 0.88 kg	E: 40.42 ± 1.06 w	NR	infectious (n = 19), hemolytic (n = 7), perinatal factors (n = 3), others (n = 2)	Probiotics + (C)	5 d	(1)(2)(3)(4)(9)(18)(21)	E: 2/31
		C: 13.68 ± 7.37 d	C: 3.52 ± 0.91 kg	C: 40.52 ± 1.09 w	NR	infectious (n = 18), hemolytic (n = 6), perinatal factors (n = 4), others (n = 3)	phototherapy			C: 4/31
Zhang (2023) [31]	74 (37/37)	E: 7.01 ± 2.02 d	NR	NR	NR	infectious (n = 15), hemolytic (n = 15), breast milk jaundice (n = 7)	Probiotics + (C)	3–10 d	(1)(2)(4)(8)(15)(17)(19)(20)(27)	E: 3/37
		C: 6.97 ± 2.13 d	NR	NR	NR	infectious (n = 13), hemolytic (n = 16), breast milk jaundice (n = 8)	phototherapy			C: 10/37
Zhang (2024) [32]	78 (39/39)	E: 8.03 ± 1.19 d	E: 3020.23 ± 1049.90 g	E: 39.86 ± 1.75 w	NR	NR	Probiotics + (C)	5 d	(1)(2)(4)(9)	E: 1/39
		C: 7.99 ± 1.21 d	C: 3020.15 ± 1049.85 g	C: 39.84 ± 1.76 w	NR	NR	phototherapy			C: 9/39
Zhu (2022) [33]	140 (69/71)	E: 2.35 ± 3.16 d	E: 3.14 ± 0.15 kg	E: 39.17 ± 1.06 w	NR	hemolytic (n = 36), hepatocellular (n = 20), cholestatic jaundice (n = 11), others (n = 2)	Probiotics + (C)	5 d	(1)(2)(4)(9)	E: 7/69
		C: 2.36 ± 2.26 d	C: 3.15 ± 0.13 kg	C: 40.12 ± 1.01 w	NR	hemolytic (n = 32), hepatocellular (n = 23), cholestatic jaundice (n = 10), others (n = 6)	phototherapy			C: 18/71

E, experimental; C, control; SD, standard deviation; NR, not reported; d, day; w, week; (1), total effective rate; (2), adverse events; (3), transcutaneous bilirubin level; (4), serum bilirubin level; (5), growth parameters: length, weight, and head circumference; (6), apolipoprotein M; (7), neuron-specific enolase; (8), C-reactive protein; (9), the time of jaundice fading (d); (10), S100B protein; (11), glial cell line-derived neurotrophic factor; (12), alanine aminotransferase; (13), aspartate aminotransferase; (14), serum albumin; (15), length of hospital stays; (16), bowel-related indicators; (17), duration of phototherapy (d); (18), serum immunoglobulin levels (immunoglobulin A, immunoglobulin G, immunoglobulin M); (19), procalcitonin; (20), interleukin-6; (21), Neonatal Behavioral Neurological Assessment score; (22), Beta-2 microglobulin; (23), gamma-glutamyl transferase; (24), 25-hydroxyvitamin D_3_; (25), lactate dehydrogenase; (26), cystatin C; (27), serum T lymphocyte; (28), intestinal flora.

**Table 2 microorganisms-13-01441-t002:** Intervention information.

First Author (Year)	Probiotics Information	Dosage (Time)	Frequency (Day)	Phototherapy Information	Frequency (Day)
Gao (2021) [14]	*C*. *butyricum* double living capsule	420 mg	2 times	NR	NR
Huang (2023) [15]	*C*. *butyricum*–*Bifidobacterium* probiotic supplementation	500 mg	2 times	Wavelength: 427 to 475 nm	NR
Lai (2020) [16]	*C. butyricum*–based triple probiotic supplementation: consisting of *C*. *butyricum*, *Enterococcus faecalis*, and *Bacillus mesentericus*	100 mg	3 times	Wavelength: 427 to 475 nm Treatment duration: 18 h	1 time
Li (2020a) [17]	*C*. *butyricum*–*Bifidobacterium* probiotic supplementation	500 mg	2 times	Treatment duration: 2 to 6 h Pause interval: 2 to 4 h Total daily exposure: less than 10 h	2 times
Li (2020b) [18]	*C. butyricum* probiotic supplementation	500 mg	3 times	Wavelength: 427 to 475 nm Treatment duration: 6 to 8 h	1 time
Li (2024) [19]	*C*. *butyricum*–*Bifidobacterium* probiotic supplementation	NR	NR	Wavelength: 420 nm Treatment duration: 12 h Pause interval: 12 h	1 time
Lin (2022) [20]	*C*. *butyricum*–based triple probiotic supplementation: consisting of *C*. *butyricum*, *Enterococcus faecalis*, and *Bacillus mesentericus*	200 mg	3 times	Total daily exposure: less than 18 h Pause interval: 6 h	NR
Liu (2024) [21]	*C*. *butyricum*–*Bifidobacterium* probiotic supplementation	NR	NR	Wavelength: 420 to 480 nm Treatment duration: 8 h Pause interval: 8 h	NR
Ren (2020) [22]	*C*. *butyricum*–*Bifidobacterium* probiotic supplementation	500 mg	2 times	Wavelength: 420 to 470 nm Treatment duration: 12 h	NR
Shi (2022) [23]	*C*. *butyricum*–*Bifidobacterium* probiotic supplementation	500 mg	3 times	Wavelength: 420 to 470 nm Treatment duration: 14 h Pause interval: 10 h	NR
Song (2024) [24]	*C*. *butyricum*–*Bifidobacterium* probiotic supplementation	500 mg	2 times	Wavelength: 425 to 475 nm Treatment duration: 3 h	2 times
Sun (2024) [25]	*C*. *butyricum*–*Bifidobacterium* probiotic supplementation	500 mg	1 time	Treatment duration: 4 h Pause interval: 5 h Total daily exposure: Less than 16 h	NR
Wang (2023a) [26]	*C*. *butyricum*–*Bifidobacterium* probiotic supplementation	400 mg	3 times	Treatment duration: 8 h Pause interval: 16 h	NR
Wang (2023b) [27]	*C*. *butyricum*–*Bifidobacterium* probiotic supplementation	400 mg	2 times	Wavelength: 420 to 470 nm Treatment duration: 12 h Pause interval: 12 h	1 time
Wang (2023c) [28]	*C. butyricum* probiotic supplementation	500 mg	2 times	Treatment duration: 8 to 12 h	NR
Xiong (2020) [29]	*C. butyricum* probiotic supplementation	500 mg	2 times	Treatment duration: 4 times/h	4 times
Zhao (2020) [30]	*C*. *butyricum*–*Bifidobacterium* probiotic supplementation	500 mg	2 times	Treatment duration: 8 h Pause interval: 4 h	2 times
Zhang (2023) [31]	*C*. *butyricum*–*Bifidobacterium* probiotic supplementation	500 mg	2 times	Wavelength: 425 to 475 nm Total daily exposure: less than 8 to 12 h	NR
Zhang (2024) [32]	*C. butyricum* probiotic supplementation	500 mg	2 times	Treatment duration: 5 to 6 h Pause interval: 2 to 4 h Total daily exposure: less than 8 to 12 h	2 times
Zhu (2022) [33]	*C*. *butyricum*–*Bifidobacterium* probiotic supplementation	500 mg	2 times	Treatment duration: 8 h	2 times

NR, not reported; h, hour.

**Table 3 microorganisms-13-01441-t003:** The quality of evidence.

Outcomes	No. Participants (Studies)	Anticipated Absolute Effects (95% CI)		Relative Effect (95% CI)	Heterogeneity (I^2^)	Quality of Evidence (GRADE)
		Risk with Control Group	Risk with Intervention Group			
Total bilirubin levels	870 (10 RCTs)	-	SMD 1.54 lower (2.21 lower to 0.86 lower)	-	95	⨁⨁⨁◯ Moderate ^a,b^
Direct bilirubin levels	410 (4 RCTs)	-	SMD 0.91 lower (1.32 lower to 0.50 lower)	-	73	⨁⨁⨁◯ Moderate ^a,b^
Indirect bilirubin levels	448 (6 RCTs)	-	SMD 2.03 lower (2.98 lower to 1.07 lower)	-	94	⨁⨁⨁◯ Moderate ^a,b^
The time of resolution of jaundice	636 (6 RCTs)	-	MD 1.2 lower (1.66 lower to 0.75 lower)	-	95	⨁⨁⨁◯ Moderate ^a,b^
Total effective rate	1679 (18 RCTs)	776 per 1000	931 per 1000 (893 to 970)	RR 1.20 (1.15 to 1.25)	45	⨁⨁⨁◯ Moderate ^a^
Adverse events	1737 (19 RCTs)	147 per 1000	57 per 1000 (43 to 78)	RR 0.39 (0.29 to 0.53)	0	⨁⨁⨁◯ Moderate ^a^
Transcutaneous bilirubin levels	502 (5 RCTs)	-	SMD 1.11 lower (1.62 lower to 0.59 lower)	-	85	⨁⨁◯◯ Low ^a,b,c^
Length of hospital stay	446 (4 RCTs)	-	MD 1.66 lower (2.44 lower to 0.88 lower)	-	94	⨁⨁◯◯ Low ^a,b,c^

^a^, The overall bias was unclear in half or more of the studies; ^b^, although substantial heterogeneity was observed (I^2^ > 50%), all studies demonstrated effect estimates in the same direction, and the variability was deemed clinically acceptable. Thus, no downgrade was employed for inconsistency; ^c^, the sample size did not satisfy the OIS criterion; CI, confidence interval; RR, risk ratio; MD, mean difference; SMD, standardized mean difference; RCT, randomized controlled trial; GRADE, Grading of Recommendations Assessment, Development, and Evaluation; OIS, optimal information size; ⊕, represents higher certainty of evidence; ◯, represents lower certainty of evidence.

## Data Availability

The original contributions presented in the study are included in the article/Appendix A, further inquiries can be directed to the corresponding authors.

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
