# Peer review of "Efficacy of *Clostridium butyricum* Supplementation Combined with Phototherapy for Neonatal Hyperbilirubinemia: A Systematic Review and Meta-Analysis"

_microorganisms, 2025, doi:10.3390/microorganisms13071441_

Round 1

Reviewer 1 Report

Comments and Suggestions for Authors

Thank you for the opportunity to review the manuscript entitled "Efficacy of Clostridium butyricum Supplementation Combined With Phototherapy for Neonatal Hyperbilirubinemia: A Systematic Review and Meta-Analysis." The authors address a clinically relevant and timely topic concerning the use of probiotics as an adjunct to phototherapy in managing neonatal hyperbilirubinemia (NH).

Overall, the manuscript is well-structured and presents a comprehensive synthesis of the available randomized controlled trials. However, I have several major concerns that need to be addressed before the manuscript can be considered for publication:

1. Although the meta-analyses demonstrate consistently favorable outcomes for the intervention group, the overall methodological quality of the included studies is suboptimal. Many studies are rated as having either high risk of bias or “some concerns,” and heterogeneity is substantial across most outcomes (I² > 90% in several analyses). This raises serious concerns about the reliability and generalizability of the pooled estimates. The limitations associated with this issue should be more explicitly discussed in both the Results and Discussion sections.

2. All included trials were conducted in China. While this reflects the available evidence, the geographic and ethnic homogeneity of the dataset greatly limits its external validity. This issue should be more clearly acknowledged and discussed, particularly in the context of differences in gut microbiota development and bilirubin metabolism across populations.

3. The manuscript refers to possible mechanisms of action (e.g., modulation of gut microbiota, β-glucuronidase inhibition), but fails to provide a critical evaluation of the direct evidence supporting these claims. The lack of mechanistic validation, particularly in neonates, may lead to overinterpretation of the results. A dedicated section summarizing the biological plausibility—along with its limitations—would help balance the interpretation of findings.

4. Given the above limitations, the authors are advised to tone down their conclusions. Statements suggesting strong clinical efficacy or recommending clinical implementation should be avoided unless supported by high-quality, multicenter data. The current form of the abstract and conclusion could mislead readers regarding the strength of evidence.

I believe the topic is important and the findings potentially valuable, but the manuscript should be revised to better reflect the limitations of the evidence. If the issues outlined above are adequately addressed, the article would be suitable for publication.

Author Response

We have diligently reviewed the comments provided by the reviewers and have carefully incorporated revisions. We have uploaded the final revised version incorporating the reviewer's comments and detailed the modifications in the attached manuscript file which are marked in red. We hope the corrections will meet with your approval.

Reviewer 2 Report

Comments and Suggestions for Authors

The research question is well-defined, and the objectives are clearly stated. but following comments need to complete to enhance the clinical significance of combining Clostridium butyricum supplementation with phototherapy

1-How bacteria are involved in improving phototherapeutic treatment outcomes… Describe in the introduction section how this bacterium is better than other bacteria.

2-Phototherapeutic treatment is needed for this bacteria. Is there any study that reported the use of this only bacteria without phototherapeutic treatment (as it is taken, the control group was treated with phototherapy alone)?

3-The age in days of neonates is not clear in the study. Is this an influencing factor?

4-What was the criteria to resolve discrepancies?

5-Signify the continuous or dichotomous statistical analysis as well as heterogeneity analysis for this study.

6-What was the value for total bilirubin levels as observed with 10 studies as mentioned in line number 237 

7-What direct bilirubin levels and total bilirubin levels conclude the significance of this study?

8-Is there any other bacteria that authors signify the conclusion of this study? 

9-what are the future Recommendations for clinical practice by this study

Author Response

(The authors gave the same response as above.)

Reviewer 3 Report

Comments and Suggestions for Authors

Strengths

  1. Well-defined Objective: The study clearly aims to assess the efficacy and safety of C. butyricum with phototherapy compared to phototherapy alone in treating neonatal hyperbilirubinemia.

  2. Systematic Methodology:

    • Comprehensive literature search across 11 databases in three languages.

    • Registration with PROSPERO (CRD420251031376) enhances transparency.

    • Followed PRISMA and GRADE standards.

  3. Robust Meta-Analysis:

    • 20 RCTs with 1,715 neonates included.

    • Results show significant improvements in bilirubin levels, jaundice resolution time, TER, and adverse event reduction.

    • Use of sensitivity and publication bias analyses to assess result stability.

Weaknesses / Concerns

  1. Limited Geographic Scope:

    • All included RCTs were conducted in China, affecting generalizability.

    • Ethnic differences in NH prevalence (as acknowledged) may limit external validity.

  2. Risk of Bias:

    • Six studies had high risk of bias; the rest had "some concerns" due to unclear allocation or reporting.

    • Some studies reported physiologically implausible bilirubin values or had missing units.

  3. Heterogeneity:

    • Substantial heterogeneity (I² > 90%) in several key outcomes (e.g., bilirubin levels, hospital stay).

    • No subgroup analysis due to insufficient study breakdown by jaundice type or probiotic strain.

  4. Reporting Gaps:

    • Lack of uniformity in phototherapy protocols and probiotic dosage/frequency across studies.

    • Some outcomes (e.g., transcutaneous bilirubin, hospital stay) had low GRADE certainty due to small sample size or methodological limitations.

Suggestions for Improvement

  • Clarify Units and Data Consistency: Studies with abnormal or missing bilirubin data should be clearly flagged and addressed or excluded.

  • Enhance Subgroup Analysis: Encourage further stratification of future data by:

    • Type of jaundice (pathological vs. physiological)

    • Probiotic formulation (single vs. multi-strain)

  • Broaden Scope: Future updates should include studies from diverse populations and countries.

  • Transparent Protocols: Detail probiotic composition, dosages, and phototherapy parameters for better reproducibility.

Comments on the Quality of English Language

minor changes for the english accuracy

Author Response

(The authors gave the same response as above.)
